# Successful Caesarean Section on Ticagrelor Treatment One Day after Primary Percutaneous Coronary Intervention

**DOI:** 10.3390/jpm13091344

**Published:** 2023-08-30

**Authors:** Nebojsa Antonijevic, Predrag Mitrovic, Nikola Gosnjic, Dejan Orlic, Sasa Kadija, Tanja Ilic Mostic, Nebojsa Savic, Ljubica Birovljev, Zaklina Lekovic, Dragan Matic

**Affiliations:** 1Clinic for Cardiology, University Clinical Center of Serbia, 11 000 Belgrade, Serbia; 2Faculty of Medicine, University of Belgrade, 11 000 Belgrade, Serbia; 3Department of Pharmacokinetics and Clinical Pharmacy, Faculty of Pharmacy, University of Belgrade, 11 000 Belgrade, Serbia; 4Clinic for Gynecology and Obstetrics, University Clinical Center of Serbia, 11 000 Belgrade, Serbia; 5Clinic for Vascular Surgery, Transfusion, University Clinical Center of Serbia, 11 000 Belgrade, Serbia

**Keywords:** ticagrelor, pregnancy, myocardial infarction, aggregometry

## Abstract

Caesarean section is a challenging intervention in patients treated with dual antiplatelet therapy. We present a case of a 32-year-old pregnant woman experiencing large acute myocardial infarction (MI) of the anterolateral wall, complicated by cardiogenic shock in the 38th week of pregnancy, and treated with drug-eluting stent implantation and dual antiplatelet therapy (DAPT) consisting of aspirin and ticagrelor. Less than 24 h after the MI delivery started, an urgent Caesarean section was indicated. As multiplate aggregometry testing showed a relatively insufficient level of ticagrelor platelet inhibition and a moderate level of aspirin platelet inhibition, a Caesarean section was performed without discontinuation of ticagrelor, which was decided due to the need for emergency surgery. Local hemostatic measures including administration of tranexamic acid were applied. The patient did not experience excessive bleeding. A healthy male baby was born. To the best of our knowledge, this is the first reported case of surgery in pregnant women treated with DAPT without ticagrelor discontinuation.

## 1. Introduction

Although acute MI in pregnancy is a rare disorder, coronary artery disease accounts for more than 20% of maternal cardiac deaths. Considering the rising age of women becoming pregnant, acute coronary syndrome might become more common [1]. Although the most common cause of myocardial infarction is coronary thrombosis, the rare cause of myocardial infarction, especially in young women, can be spontaneous dissection of the coronary artery [2].

Currently, the management of antiplatelet therapy in urgent surgeries is not extensively covered in clinical guidelines. Usually, in high-ischemic-risk patients, aspirin should not be discontinued regardless of the hemorrhagic risk of the surgery, whereas P2Y12 antagonists (ticagrelor, prasugrel, and clopidogrel) administration should be discontinued when possible [1,3]. Here, we present a complex case of a pregnant woman experiencing MI with ST-segment elevation (STEMI) and treated with DAPT consisting of aspirin and ticagrelor, without discontinuation of ticagrelor prior to Caesarean delivery.

## 2. Case Report

A 32-year-old female patient in the 38th week of gestation was admitted to the Reanimation Ambulance of the Emergency Centre in a state of cardiogenic shock and clinical signs of STEMI of the anterolateral wall. On admission, the patient was found to be dyspneic, hypotensive (arterial tension: 85/65 mmHg both-sided), and tachycardic (heart frequency: 115 per minute). Medical history was taken—she was a former smoker and hypertensive two years preceding pregnancy. Hereditary factors were considered to have strongly contributed to her state, as the patient’s pregnant sister suddenly died at the age of 35 during the ninth month of pregnancy, while her other sister suddenly died at the age of 38. In addition, the patient’s father died of MI at the age of 55.

Electrocardiography testing was performed with the results showing ST elevation in leads D1, aVL, and V2–V6 and ST depression in leads D2, D3, and aVF, as displayed in Figure 1.

Echocardiography testing showed a slightly enlarged left ventricle (5.7 cm in diastole and 3.7 cm in systole), hypokinetic medial segments of the interventricular septum, anterior and lateral walls, and akinetic apical segments. The total left ventricle ejection fraction (EF) was 30%. The Echocardiography examination also showed a normal diameter of the ascendant aorta with no signs of aortic dissection.

Laboratory analysis showed the following results: Troponin T was 98 ng/L (normal value of up to 14 ng/L) at first measure, reaching a maximum value of 10,000 ng/L; creatine kinase (CK) was 7263 U/L (normal value of up to 150 U/L); platelet count was 354 × 10^9^/L (normal range of 158–424 × 10^9^/L); D dimer was 0.84 mg/L (normal value of up to 0.5 mg/L); and the pH of the arterial blood was 7.381 (normal range of 7.350–7.450). The value of hemoglobin prior to the PCI was 87 g/L (normal range of 119–157 g/L), with hemoglobin dropping to 75 g/L, 18 h after the PCI.

The obstetric examination showed an aligned cervix with 2 cm to 3 cm dilation and regular heartbeats in the neonate. DAPT consisting of 300 mg of aspirin and 180 mg of ticagrelor, with the support of inotropic drugs, was administered. At the beginning of the PCI procedure, unfractionated heparin was given in a dose of 50 U/kg. The angiography disclosed a 90% diameter stenosis in the proximal segment of the left anterior descendent coronary artery (LAD) and left coronary sinus aneurysm. There were no signs indicating spontaneous coronary artery dissection (SCAD). Due to the high thrombus burden, bolus eptifibatide was administered. (Figure 2a)

After diagnosing single-vessel disease, a drug-eluting stent (DES, type: Premier^®^ 3.0 × 28 mm, Boston Scientific, Marlborough, MA, United States of America) was implanted in the osteo-proximal part of the LAD (Figure 2b).

Shortly after the percutaneous coronary intervention (PCI) with DES implantation, the patient was temporarily stabilized with reduced signs of left-sided heart failure (HF) but deteriorated the next day. Further obstetric examination showed cervix dilation of 4 cm, vertex presentation engaged, regular heartbeats of the baby, and a normal and adequate volume of amniotic fluid and placenta on the back wall. The Doppler obstetric ultrasound showed normal blood flow through the umbilical cord. Due to significant cervix dilation and persisting signs of HF, an urgent Caesarean section was indicated.

As a part of the perioperative assessment, laboratory analyses including aggregometry testing were performed. Impedance aggregometry testing on a Multiplate^®^ device (Roche^®^ Diagnostics, Basel, Switzerland) revealed a relatively insufficient level of ticagrelor platelet inhibition (area under the curve: 591 AU*min) in the adenosine diphosphate (ADP) high-sensitive test (ADP + prostaglandin E), while a moderate level of aspirin-induced platelet inhibition was detected (area under the curve: 404 AU*min) using the arachidonic acid (ASPI) test. The thrombin receptor activating peptide (TRAP) test, which represents baseline platelets aggregation, showed expected values of 1399 AU*min. ADP high-sensitive, ASPI, and TRAP test results can be seen in Figure 3.

Based on these aggregometry results, it was decided that the bleeding risk was minimal, with a high ischemic risk and therefore, DAPT was not interrupted prior to delivery. A Caesarean section was performed with intraoperative blood salvage, transfusion of red cell concentrates (due to a preoperative hemoglobin value of 75 g/L), and local hemostatic measures (application of six ampules of tranexamic acid via a surgical incision (3000 mg) and two ampules in the front abdominal wall muscles (1000 mg)). Due to uterine hypotonia, one ampule of Prostaglandin M 15 (250 µg) was injected into the myometrium. The procedure passed without complications, and a male baby of 2700 g weight was born with an estimated standardized assessment for infants after delivery (APGAR) score of eight.

In the subsequent treatment course, the patient had the following findings: blood in lochia secretion and mild epistaxis, paroxysmal atrial fibrillation, parietal left ventricular thrombus, bilateral bronchopneumonia, and pleural effusions. She also had anemia (without meeting the criteria of major bleeding) and psycho-organic syndrome.

Taking into consideration epistaxis and blood in lochia secretion, it was decided to replace ticagrelor with clopidogrel. However, after receiving a loading dose of 600 mg of clopidogrel in two divided doses, aggregometry testing showed even more significantly pronounced clopidogrel resistance. Therefore, clopidogrel was discontinued and ticagrelor was reintroduced. In the following course, aggregometry testing showed adequate response to ticagrelor, as shown in Table 1. Special attention was drawn to the patient regarding the prohibition of breastfeeding due to ticagrelor administration.

Following detailed thrombophilia testing, the results showed a transitory positive lupus anticoagulant LA1/LA2: 1.74, …, 1.65, …, 1.43 (normal range of 1.1–1.4), a transitory decrease in protein C global (0.45, …, 0.73, …, 1.10 µmol/L; normal levels of protein C 3.9–5.9 µmol/L), transitory elevated factor VIII (>145%), and elevated homocysteine (17.4 µmol/L; normal range of 0–14 µmol/L). On genetic examination, we detected MTHFR C677T heterozygote mutation, PAI-1 4G/5G heterozygote mutation with normal FII 20210: GG normal and normal FV Leiden. The antithrombin III level was normal (94%; normal range of 79–112%). The value of the C reactive protein was mildly elevated, slightly varying during hospitalization, and did not indicate arteritis.

Anti-cardiolipin antibodies showed normal values.

Analysis of the lipidogram revealed that the total cholesterol was in the normal range, but high-density lipoprotein (HDL) cholesterol was decreased (0.78 mmol/L, normal value above 1.3 mmol/L. Triglycerides were elevated, ranging from 3.29 to 1.75 mmol/L (normal values of up to 1.7 mmol/L), as well as elevated Lipoprotein (a) (0.64 g/L, normal range of 0–0.3 g/L).

After leaving the intensive care unit, the patient’s antithrombotic therapy included aspirin (100 mg), ticagrelor (90 mg twice daily), and enoxaparin 0.6 mL subcutaneously. On discharge, enoxaparin was replaced with warfarin. The patient’s CHA_2_DS_2_-VASc score for stroke risk assessment in atrial fibrillation at discharge was estimated to be a 3. The value of ≥3 for women represents “moderate-high” risk and anticoagulant therapy is recommended. The triple antithrombotic therapy was provided due to the high thrombotic risk caused by an enlarged zone of myocardial infarction with the presence of a huge akinetic area and paroxysmal atrial fibrillation. Warfarin instead of NOAK was provided due to the lower social status of the patient.

Three years after the surgery, both the mother and the baby were in stable condition.

## 3. Discussion

Pharmacological treatment of acute MI in pregnancy represents a clinical challenge in which both pregnant women and the baby’s safety must be considered. Low-dose aspirin is considered generally safe, whereas information on P2Y12 antagonists is insufficient [1]. However, taking into account the hypercoagulable nature of the pregnancy [4,5] and the platelet aggregation changes [6,7], individualized approaches may prove especially useful. In the current literature, only two case reports of the use of ticagrelor in pregnant women with MI were described. In the first case, ticagrelor was discontinued two weeks before delivery [8], whereas in the second case, ticagrelor was stopped five days prior to the procedure, with tirofiban infusion antiplatelet therapy bridging [9]. Similar cases describing prasugrel use in pregnancy and its discontinuation prior to delivery are described in the literature [10,11]. Also, clopidogrel was discontinued in most cases available in the literature five to seven days prior to delivery or earlier [12,13,14]. We were able to identify two cases when the Caesarean section was performed without clopidogrel discontinuation. In the first case, platelet transfusion was administered and no aggregometry testing was performed [15]. In the other case, clopidogrel was withheld on the morning of the urgent Caesarean section and platelet function testing was not reported [16].

Our patient required an emergency Caesarean section. Based on the aggregometry results and our previous experience with platelet function testing, the decision was made to perform the procedure without the discontinuation of ticagrelor. However, we emphasize the importance of using systemic and local hemostatic preventive measures—intraoperative blood salvage with the local use of tranexamic acid.

According to the manufacturer, normal values for the multiplate ADP high-sensitivity test (Multiplate^®^, Roche Diagnostics, Basel, Switzerland) range from 430 AU*min to 1000 AU*min. Although current guidelines recommend the discontinuation of ticagrelor at least five days prior to surgery [1,17], a recent retrospective study showed five days to be possibly longer than necessary—i.e., there was no significant increase in the major bleeding risk in acute coronary syndrome patients operated on 72–120 h after ticagrelor discontinuation, as compared to those operated on more than 120 h after the discontinuation [18].

C. J. Malm et al. performed a prospective observational study in ticagrelor-treated cardiac surgery patients. The authors found the optimal cut-off value for ADP-induced aggregation to be 220 AU*min. According to their study, 61% of patients developed severe bleeding when ADP-induced aggregation was below the cut-off value, compared to only 14% of patients when aggregation was at or above the cut-off value [19]. Ranucci et al. suggested a cut-off of 310 AU*min for the Multiplate ADP test to support a clinical decision to postpone elective surgery [20]. Similar to the aforementioned results, Sebastian Woźniak et al. found that in patients undergoing coronary artery bypass graft surgery, the results of the ADP test (multiple electrode platelet aggregometry) of less than 260 AU*min strongly predicted serious bleeding complications after coronary artery bypass graft surgery [21]. With the value of 591 AU*min for ADP-induced aggregation in our patient, her results were well above the aforementioned cut-off values.

A large meta-analysis in 4213 patients showed that the use of platelet reactivity testing after PCI and consequent appropriate intensification of antiplatelet therapy reduces cardiovascular mortality and stent thrombosis following PCI [22,23].

Since systemic use of antifibrinolytics may increase the ischemic risk but decrease major bleeding [24], we opted for the local use of tranexamic acid. According to the guide of an Italian group of experts, aspirin should be continued even if the risk of bleeding is high, whereas the approach with P2Y12 antagonists differs based on the type of surgery. Although the Caesarean section was not described, the authors of the guideline suggested that individual bleeding and ischemic risk assessment should be performed. A multidisciplinary approach is to be taken to tailor adequate antiplatelet therapy in high-risk patients needing the surgery [3].

The relationship between antithrombotic therapy, bleeding, and ischemic events is extremely complex. Continuous efforts are being made to find new therapeutic approaches in primary PCI that would minimize the risk of bleeding without increasing the risk of thrombotic complications. These risks are of great importance, especially when surgical intervention needs to be performed immediately after the primary PCI. An effort to suppress important ischemic events while trying to steer clear of major bleeding complications was described brilliantly by Deepak Bhatt. He compared the wily Odysseus in Greek mythology who successfully navigated between a ferocious beast Scylla and a monstrous whirlpool Charybdis to an astute clinician managing myocardial ischemia by using antiplatelet therapy who attempts to balance coronary thrombosis, the basic cause of myocardial ischemia, against hemorrhage, the most feared complication of antiplatelet therapy [25]. 

Acute coronary syndrome and pregnancy are becoming more commonly associated, especially since pregnancy is more common in the fourth decade of life or later. It is necessary to carefully assess the risk of thrombosis versus the risk of bleeding according to the individual characteristics of each single patient. The treatment of these patients requires individualized antiplatelet therapy based on point-of-care testing of platelet function. Our case shows that taking into account a patient’s response to antiplatelet therapy, it is possible to successfully treat acute myocardial infarction with primary PCI and perform emergency surgical intervention very soon after that, such as urgent Caesarean section in a woman on dual antiplatelet therapy. 

## 4. Conclusions

To the best of our knowledge, this is the first case of pregnancy-associated MI treated with DAPT without ticagrelor discontinuation prior to delivery. In surgeries, aspirin is regarded as a relatively safe and possibly useful drug when used in small doses. On the other hand, evidence supporting decisions on ticagrelor and other P2Y12 antagonists in urgent surgeries is lacking. Therefore, laboratory monitoring of platelet aggregation may provide invaluable insight into the overall bleeding and ischemic risk. However, the use of antiplatelet therapy in pregnancy and delivery requires significant pharmacotherapeutic considerations, and a multidisciplinary approach should always be applied.

## Figures and Tables

**Figure 1 jpm-13-01344-f001:**
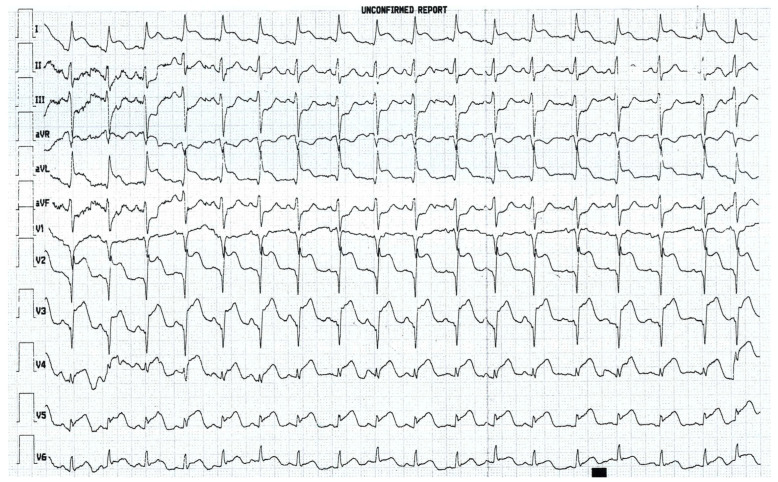
ECG on admission.

**Figure 2 jpm-13-01344-f002:**
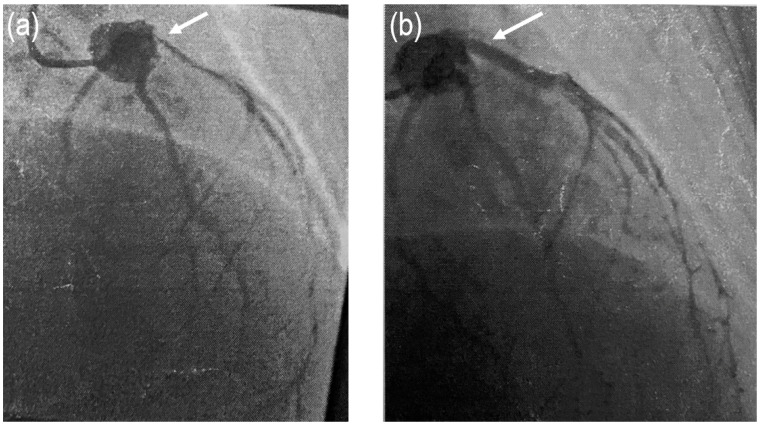
Coronary angiography: (**a**) stenosis of the proximal segment of left anterior descendent coronary artery (arrow). (**b**) Recovery of coronary flow after stent implantation (arrow).

**Figure 3 jpm-13-01344-f003:**
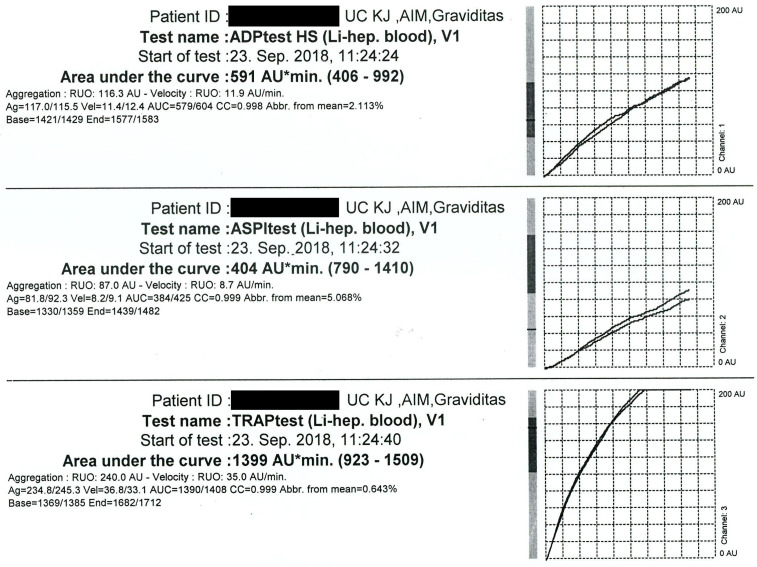
Impedance aggregometry testing results.

**Table 1 jpm-13-01344-t001:** Aggregometry findings during hospital stay.

Hospitalization Day	ADP Inhibitor Drug	ADP Test HS (AU*min)	ASPI Test (AU*min)	TRAP Test (AU*min)
Day 1	ticagrelor	591	404	1399
Day 2	clopidogrel	1081	889	1984
Day 3	clopidogrel	1263	1109	1718
Day 4	ticagrelor	567	1320	1807
Day 5	ticagrelor	535	646	1679
Day 6	ticagrelor	373	373	1761
Day 9	ticagrelor	315	512	1352

ADP—adenosine diphosphate; HS—high sensitive; AU—area under the curve; ASPI—arachidonic acid; TRAP—Thrombin receptor activating peptide.

## Data Availability

The data presented in this study are available upon request from the corresponding authors.

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
