# Peer review of "Successful Caesarean Section on Ticagrelor Treatment One Day after Primary Percutaneous Coronary Intervention"

_jpm, 2023, doi:10.3390/jpm13091344_

Round 1
Reviewer 1 Report
The case is interesting. Little data in the literature. Guidelines do not show a clear pathway of what to do with DAPT in pregnant women with ACS.
Useful data on assessing platelet function and resistance to DAPT provided that the tests performed are validated and have clear cut-off points for each test.
Please add that it was possible to:
1- Differential diagnosis: aortic dissection (may have dissected the trunk of the left coronary artery and given a picture of infarction. Differential all the more important because the patient's family history may suggest aortic aneurysms in her sisters and mother.
2- differential diagnosis spontaneous dissection of the coronary artery SCAD- mainly in women, hormonal factor, often in the perinatal period- in this case, stents are not implanted rather
It should be entered that the pavjent was banned from feeding due to taking ticagrelor.
please enter CHADsVASc- patient received DAPT and anticoagulant at exit
Please complete the literature on SCAD:
Spontaneous dissection as a rare cause of infarction in young women.
Klotzka A, Iwańczyk S, Barczynski M, Araszkiewicz A, Mitkowski P, Lesiak M.
Kardiol Pol. 2021;79(11):1274-1275. doi: 10.33963/KP.a2021.0060. epub 2021 Jul 16
Author Response
The response to reviewer 1 comments is contained in the file below.

Reviewer 2 Report
The authors present an extremely complicated case of 32-year pregnant women in the 38th week of pregnancy who had large acute myocardial infarction of anterior wall complicated by cardiogenic shock with an indication for Caesarean section within the first 24 hours of infarct presentation. The patient appears to have a strong familial history for cardiovascular disease and apparently resistance to antithrombotic drugs. The case is extremely well presented in terms of the amount of detailed information provided by the authors. I have no major criticisms with respect to handling of this very difficult case and the successful outcome speaks for itself. I have only minor comments and requests for the authors:
1. Can the authors summarize their experience with this case in order to deal with other future similar cases? This could be shortly discussion in discussion section.
2. Was unfractionated heparin (not mentioned) or other periprocedural anticoagulants used in the setting of primary PCI procedure?
3. The authors state that patient was discharged on aspirin, ticagrelor and warfarin. This regimen was unusual and extremely dangerous with respect to bleeding risk. What was the indication for warfarin. Why the authors did not prefer rivaroxaban or apixaban or other NOAC? They stated that atrial fibrillation was paroxysmal.
4. Did the authors consider cangrelor instead of ticagrelor (as bridging to Caesarian surgery)?
5. I advise the authors to change the discussion to focus more on their case, justify their therapeutic approach, discuss the brittle equilibrium between bleeding and ischemic risk in this very specific case than discuss differences between aggregation tests.
6. Detailed lipidogram analysis is not particularly helpful in the setting of this study. Platelet count, D-dimers and hemoglobin drop as well as acid-base balance parameters could have been more important.
Minor edition needed.
Author Response
The response to reviewer 2 comments is contained in the file below.
